# Therapeutic Potential of Orally Administered Rubiscolin-6

**DOI:** 10.3390/ijms24129959

**Published:** 2023-06-09

**Authors:** Yusuke Karasawa, Kanako Miyano, Masahiro Yamaguchi, Miki Nonaka, Keisuke Yamaguchi, Masako Iseki, Izumi Kawagoe, Yasuhito Uezono

**Affiliations:** 1Department of Pain Medicine, Juntendo University Graduate School of Medicine, Tokyo 113-8421, Japan; yusuke.karasawa@viatris.com (Y.K.); kmiyano@ncc.go.jp (K.M.); masahiro.yamaguchi@pfizer.com (M.Y.); keisuke@juntendo.ac.jp (K.Y.); miseki@juntendo.ac.jp (M.I.); 2Department of Pain Control Research, The Jikei University School of Medicine, Tokyo 105-8461, Japan; minonaka@jikei.ac.jp; 3Medical Affairs, Viatris Pharmaceuticals Japan Inc., Tokyo 105-0001, Japan; 4Division of Cancer Pathophysiology, National Cancer Center Research Institute, Tokyo 104-0045, Japan; 5Medical Affairs, Pfizer Japan Inc., Tokyo 151-8589, Japan; 6Department of Anesthesiology and Pain Medicine, Faculty of Medicine, Juntendo University, Tokyo 113-8421, Japan; ikawago@juntendo.ac.jp; 7Supportive and Palliative Care Research Support Office, National Cancer Center Hospital East, Chiba 277-8577, Japan

**Keywords:** spinach, rubiscolin, peptide, opioid, delta-opioid receptor

## Abstract

Rubiscolins are naturally occurring opioid peptides derived from the enzymatic digestion of the ribulose bisphosphate carboxylase/oxygenase protein in spinach leaves. They are classified into two subtypes based on amino acid sequence, namely rubiscolin-5 and rubiscolin-6. In vitro studies have determined rubiscolins as G protein-biased delta-opioid receptor agonists, and in vivo studies have demonstrated that they exert several beneficial effects via the central nervous system. The most unique and attractive advantage of rubiscolin-6 over other oligopeptides is its oral availability. Therefore, it can be considered a promising candidate for the development of a novel and safe drug. In this review, we show the therapeutic potential of rubiscolin-6, mainly focusing on its effects when orally administered based on available evidence. Additionally, we present a hypothesis for the pharmacokinetics of rubiscolin-6, focusing on its absorption in the intestinal tract and ability to cross the blood–brain barrier.

## 1. Introduction

Proteins play crucial roles in the human body, serving as enzymes, hormones, neurotransmitters, and bone and muscle components. Several proteins or peptides exert strong bioactivities even when they are present in small amounts; therefore, they may be used as therapeutic agents. Currently, advancements in biotechnology, biologics, and proteins, such as antibodies and hormones, have been developed and exploited in clinical practice [1]. From the perspective of drug development, the specificity of biologics to target sites is beneficial because it makes them safer to use than traditional non-specific drugs consisting of small molecules. However, there are a few disadvantages of biologics, such as in vivo instability, low permeability into the intestinal tract, and expensive manufacturing and quality control processes.

In recent years, naturally occurring or chemically synthesized peptides have been considered for novel drug development, including opioid peptides [2,3]. The approaches for developing a peptide drug are based on the expectation that peptides will be more effective, considering their higher specificity and lower toxicity than small molecules, which are broadly distributed in the body. Moreover, peptides have the advantages of both small molecule and biological drugs, including reduced running cost and increased specificity to the target site [4]. Nevertheless, the method of administration of peptide drugs is basically restricted to injection, similar to that of biologics. Generally, orally ingested proteins or oligopeptides are thought to be rapidly disassembled by digestive or proteolytic enzymes, and owing to their high polarity and molecular weights, they rarely penetrate the gastrointestinal mucosa while maintaining their original structure [5]. Therefore, peptide drugs become inactive when orally administered. Given that injection is not preferred by patients because of pain at the injection site, treatment with peptide drug injection could possibly result in treatment termination because of poor adherence. To address these issues, several approaches to enhance permeability, chemically modify the molecular structure, and exploit novel drug delivery systems have been explored [6]. However, the development of a peptide drug that is active after oral administration has not been achieved.

Rubiscolins, which are composed of penta- or hexapeptide (Tyr-Pro-Leu-Asp-Leu and Tyr-Pro-Leu-Asp-Leu-Phe), are spinach-derived naturally occurring oligopeptides produced by pepsin digestion of _D_-ribulose-1,5-bisphosphate carboxylase/oxygenase (RuBisCO), the major enzyme involved in carbon dioxide fixation and photorespiration [7]. The pharmacological properties of rubiscolins as delta-opioid receptor (DOR) agonists have been elucidated [7,8,9,10,11,12,13,14]. In general, the DOR is responsible for regulating a wide range of physiological functions and is associated with disease symptoms, including persistent pain, emotional disorders, and neurological disorders [15,16]. To date, none of the DOR agonists have been approved for clinical use by regulatory bodies, unlike mu-opioid receptor (MOR) agonists, which have been widely utilized for a long time for relieving pain related to various pathologies [17]. However, MOR agonists are occasionally associated with severe adverse effects, such as sedation, respiratory depression, and tolerance. Therefore, novel and safe options are urgently needed to replace existing opioids. We believe that rubiscolin-6 is a potential candidate for novel medication development.

There are two subtypes of rubiscolins, rubiscolin-6, which evokes the DOR agonistic effect equivalent to that evoked by SNC-80, an existing specific DOR agonist, and rubiscolin-5, which has been demonstrated to be relatively weaker than both SNC-80 and rubiscolin-6 [18]. The robust efficacy of rubiscolin-6 has been investigated in several in vivo studies using animal models mimicking a variety of disease states [7,8,11,12,13,14]. In addition, rubiscolins have an attractive property of oral availability. Furthermore, rubiscolin-6 is a G protein-biased specific DOR full agonist that is not activated in the β-arrestin-mediated pathway upon binding to any type of ORs, indicating that it could be safer than available opioids. Additionally, the limited effects on endogenous ligands or opioid analgesics activating MOR or kappa-OR (KOR) could enable its use with an MOR agonist when the total amount of MOR agonist used has to be reduced. Considering the adverse effects of available opioids and the circumstances of the “opioid crisis” [19], it is noteworthy that rubiscolin-6 could be a potentially novel and safer drug than MOR agonists as it evokes pain-relieving effects and various central effects via DOR [18]. Here, we review the preclinical studies of rubiscolins to elucidate their therapeutic potential from the perspective of drug development.

## 2. Therapeutic Potential of Rubiscolins

### 2.1. Orally Administered Rubiscolins

Several effects of orally administered rubiscolins observed in animal studies are shown in Table 1.

#### 2.1.1. Antinociceptive Effect

Rubiscolins were initially expected to exhibit analgesic activity, similar to other opioid compounds. A previous study revealed that orally administered rubiscolins have antinociceptive effects [7]. While rubiscolin-6 exhibited the effect at a dose of 100 mg/kg, at least 300 mg/kg rubiscolin-5 was needed to observe the same effect, indicating that rubiscolin-6 was three times more potent than rubiscolin-5. Given that the effect was blocked by intracerebroventricularly (i.c.v.) injected naltrindole, a specific DOR antagonist, the effect was mediated via the DOR pathway in the central nervous system (CNS). To date, rubiscolins are the only naturally occurring oligopeptides that have been shown to exert analgesic effects after oral administration.

#### 2.1.2. Memory-Enhancing Effect

Research using a step-through type passive avoidance test has demonstrated that rubiscolin-6 at a dose of 100 mg/kg enhances memory consolidation [8]. In contrast, rubiscolin-5 showed no significant effect by not only per os (p.o.) administration but also i.c.v. administration. The effect of rubiscolin-6 was blocked by i.c.v. naltrindole, suggesting that the effect involved the central DOR pathway.

#### 2.1.3. Anxiolytic Effect

Rubiscolin-6 at a dose of 100 mg/kg showed anxiolytic activity in an elevated plus-maze test [11]. The activity was blocked by i.c.v. naltrindole and a dopamine D1 antagonist, SCH23390, but not by a dopamine D2 antagonist, raclopride. The effect was also blocked by the sigma 1 receptor antagonists BMY14802 and BD1047. These results suggest that the anxiolytic effect of rubiscolin-6 is mediated by the dopamine D1 and sigma1 receptors downstream of DOR.

#### 2.1.4. Orexigenic and Anorexigenic Effects

A modulating effect of rubiscolin-6 on food intake has been reported [12,13,14]. First, rubiscolin-6 at doses of 0.3–1.0 mg/kg stimulated food intake, but this effect was blocked by i.c.v. naltrindole [12]. The orexigenic effect was also inhibited by intraperitoneally administered celecoxib, a cyclooxygenase (COX)-2 inhibitor, but not by intraperitoneally administered SC-560, a COX-1 inhibitor, suggesting that the effect was mediated by COX-2. In addition, leptomeningeal lipocalin-type prostaglandin D synthase (L-PGDS) in addition to the DP1 and Y1 receptors was involved in the pathway. Another study showed that rubiscolin-6 at a dose of 1 mg/kg stimulated food intake in both young and aged mice (2–27 months old), although the aged mice were resistant to ghrelin, an oligopeptide known to stimulate food intake in animals and humans via hypothalamic neuropeptide Y-mediated signaling [14]. Conversely, the anorexigenic effect of rubiscolin-6 at a dose of 0.3–1.0 mg/kg was observed in mice on a high-fat diet [13]. This effect was blocked by i.c.v. administered naltrindole as well as by i.c.v. administered HS024, an antagonist for the melanocortin 4 receptor. Taken together with the results of additional tests using wild-type and L-PGDS knock-out mice, the effect of rubiscolin-6 is mediated via pathways downstream of the central DOR.

### 2.2. In Vivo Oligopeptide Transportation

Evidence for the pharmacokinetics of rubiscolins remains scarce, despite their attractive property of oral availability. Generally, orally administered opioid peptides are inactive owing to rapid degradation into dipeptides or tripeptides by endogenous peptidases in the process of digestion before absorption. However, it is likely that some amount of rubiscolin-6 escapes the system. Stefanucci et al., investigated the in vitro intestinal stability and bioavailability of rubiscolin-6 using single layers of CaCo2 cells as a model of absorption in the small intestine [20]. They reported that approximately 10% of rubiscolin-6 was transepithelially transported. The mechanism by which rubiscolin-6 crosses the blood–brain barrier (BBB) is also unknown. Hence, two pathways should be elucidated to understand the pharmacokinetics of rubiscolin-6, which can activate DOR via oral administration: first, the mechanism by which rubiscolin-6 is absorbed while maintaining its original structure; second, the mechanism by which rubiscolin-6 can penetrate the BBB to evoke central effects.

Regarding the first mechanism, several previous studies have focused on transporters related to the absorption of oligopeptides in the intestinal tract, especially sodium-coupled oligopeptide transporters (SOPTs), which are of two subtypes, namely SOPT-1 and SOPT-2; these SOPTs are involved in oligopeptide transportation [21]. They are distinct from traditional peptide transporters known as peptide transporter-1 (PEPT-1) or PEPT-2 [22], which only transport dipeptides and tripeptides. SOPT-1 and SOPT-2 are expressed in the small intestinal epithelium, as well as in retinal pigment epithelial cells and neurons [21,23,24]. In the intestinal cells, they transport endogenous, exogenous, and synthetic oligopeptides, which consist of five or more amino acid residues, including Tat47–57, a fragment of the Tat protein encoded by human immunodeficiency virus-1, and DADLE ([_D_-Ala2, _D_-Leu5]-enkephalin), a selective DOR agonist [21]. Among the two subtypes of SOPT, SOPT-2 is considered to be involved in transporting an opioid peptide as a substrate.

Regarding the second mechanism, rubiscolin-6 has been considered to enter the CNS via BBB, although the mechanism of opioid peptide drug delivery to the CNS remains unclear [25]. Four types of peptide transporters, namely peptide transport system (PTS)-1, PTS-2, PTS-3, and PTS-4, play a role in transporting oligopeptides in brain microvascular endothelial cells. Among them, PTS-1 has been reported to be involved in the penetration of opioid peptides [26,27]. However, the detailed mechanism has not yet been elucidated. In a study on the intravenous injection method, endogenous opioid peptides, such as Tyr-MIF-1 (Tyr-Pro-Leu-Gly-NH_2_), Met-enkephalin (Tyr-Gly-Gly-Phe-Met), Leu-enkephalin (Tyr-Gly-Gly-Phe-Leu), and β-casomorphin (Tyr-Pro-Phe-Pro-Gly-Pro-Ile), were transported by PTS-1, whereas β-endorphin (31 amino acid residues), kyotorphin (Tyr-Arg), dermorphin (Tyr-d-Ala-Phe-Gly-Tyr-Pro-Ser), and morphiceptin (Tyr-Pro-Phe-Pro) were not. PTS-1 demonstrated the ability to transport opioid oligopeptides with five or more amino acid residues, but it did not transport dipeptide kyotorphin and dermorphin, which have a seven amino acid sequence, similar to that of β-casomorphin. Accordingly, it may not be easy to determine the transportation of peptides by considering only the number of amino acid residues. Although the detailed mechanism remains unknown, factors, such as size, structure, polarity, charge, and the protein binding rate of a peptide are considered to be related to the selectivity for transportation by a transport system [28].

## 3. Discussion

### 3.1. Therapeutic Potential of Orally Administered Rubiscolin-6

DOR is abundantly distributed in the limbic system, which comprises the hippocampus, amygdala, and medial prefrontal cortex, and plays a critical role in mood modulation. Therefore, drugs that target DOR are useful for not only relieving pain but also treating psychological disorders. As rubiscolin-6 has shown the potency to improve a variety of symptoms related to central functions via DOR, it is a promising candidate for drug development. Considering the various effects of p.o. rubiscolin-6, it could meet the needs in therapeutic areas as a safe opioid, the first treatment for memory consolidation, a novel mood modulator with a new mechanism of action, and a unique food intake modulator with both anorexigenic and orexigenic effects.

There is a need for an alternative option for pain relief, considering the ongoing opioid epidemic [19]. Although opioid analgesics, such as morphine, fentanyl, oxycodone, and hydrocodone, have been clinically utilized, their administration is sometimes hampered by adverse effects, which can occasionally be fatal [29]. In addition, available opioids can lead to tolerance as a result of long-term administration [30], resulting in increased opioid dosage, which can lead to severe adverse effects and eventually force physicians to switch therapy [31]. One of the mechanisms of the onset of opioid tolerance is the desensitization of OR triggered by the internalization of OR from the cell membrane into the cytoplasm. Given that internalization is related to the capture of OR by β-arrestin [32], it is preferable for opioid analgesics to be less susceptible to the protein. In this regard, it has already been revealed that rubiscolin-6 rarely activates the β-arrestin-mediated pathway upon binding to any type of ORs [18]. As rubiscolin-6 is a G protein-biased DOR agonist and does not interfere with the endogenous or exogenous ligands activating MOR and KOR [18], it can be a safer selective DOR agonist than other available opioid compounds. However, it should be noted that DOR agonists are relatively weaker in terms of antinociception than MOR agonists [33]. Therefore, rubiscolin-6 would be a safe but mild analgesic. Approximately a three-fold dose of rubiscolin-5 is needed to exert the same effect as a single dose of rubiscolin-6; this may be due to the last position of Phe in rubiscolin-6, which is critical for exerting the antinociceptive effect. In addition to opioid-related problems, neuropathic pain is clinically considered another important issue in chronic pain treatment. Neuropathic pain is reportedly long lasting and more severe than other types of chronic pain and is thus associated with a decreased quality of life [34] and increased medical costs [35]. It is defined as “pain caused by lesion or disease of the somatosensory nervous system” by the International Association for the Study of Pain. Diseases resulting from nutritional metabolism, genetic factors, trauma, ischemic conditions, toxicity, and infections are associated with neuropathic pain. Even though there are no studies on the efficacy of rubiscolin-6 for neuropathic pain, SNC-80, a representative DOR agonist, suppressed mechanical allodynia and injury-induced TNF-α upregulation in the sciatic nerve of a CCI rat model of neuropathic pain [36]. Given that DOR is reportedly involved in the modulation of neuropathic pain [37,38], future studies should elucidate the effect of rubiscolin-6 on neuropathic pain.

There are no approved drugs for memory consolidation. The role of opioids in learning and memory is not yet fully understood. Previous studies have reported that the memory-consolidation effect is mediated by the MOR, DOR, and KOR pathways. Different results were obtained, such as impairment by the MOR agonist [39], amelioration of impairment by the KOR agonist [40], and controversial results pertaining to the DOR agonist [41,42]. Rubiscolin-6 reportedly improved memory consolidation in animal studies [6]. Similar effects have been exerted by gluten exorphin A5 (Gly-Tyr-Tyr-Pro-Thr), a food-derived peptide selective for DOR, suggesting that the effect may be common among the selective DOR agonists [43]. Given that the dose of rubiscolin-6 needed for the effect was one-third of the dose of gluten exorphin A5, rubiscolin-6 may have a more potent memory-enhancing effect. However, memory impairment is one of the general symptoms that decrease the quality of life rather than a specific disease; thus, it may be suitable for rubiscolin-6 to be developed as a supplement rather than as a prescription drug with the indication.

A novel mood modulator is desired because approximately 5% of adults suffer from depression worldwide [44]. Among available treatments for emotional disorders, monoamine modulators, such as selective serotonin- and serotonin-noradrenalin-reuptake inhibitors, are prescribed as first-line treatment for major depressive disorders, but they generally require at least 1 month to exhibit their effects and are sometimes insufficient for clinical remission. Benzodiazepines, prescribed for relieving anxiety, exhibit serious adverse effects, including psychological and physical dependence [45]. Therefore, an alternative option that has a different mode of action is urgently required. We believe that DOR agonists can be candidates as DOR plays an important role in modulating moods, such as anxiety and depression [46], while MOR possibly promotes anxiety and depressive behavior [47].

Orally administered rubiscolin-6 has been shown to exert anxiolytic activity. In addition, rubiscolin-6 showed an antidepressant-like effect in a study using restraint-stressed mice, although it was not administered p.o. (it was administered intraperitoneally) [48]. Research on the critical role of DOR in depression and anxiety [49,50,51] has resulted in the identification of NC-2800, one of the DOR agonists, to be effective as an emotional modulator. Currently, NC-2800 is under phase 1 clinical testing for utility against major depressive disorder (https://jrct.niph.go.jp/en-latest-detail/jRCT2071210033; accessed on 20 November 2022). Even though the detailed pharmacological profiles of NC-2800 are not fully known (not published), rubiscolin-6 has an advantage over NC-2800; it is rarely affected by β-arrestin [18]. The most strategic scenario for developing rubiscolin-6 as a treatment option may be as an anxiolytic drug. Furthermore, rubiscolin-6 is expected to be an ideal drug for chronic pain if it can simultaneously exert analgesia and mood modulation, as chronic pain is often modified by a patient’s psychological background and related factors [52].

There is scope for developing a food intake modulator based on two requirements: to increase appetite in patients with cancer or vulnerable elderly individuals and to decrease the dietary intake of obese individuals. Interestingly, rubiscolin-6 could be utilized for the two opposite therapeutic indications and is an ideal candidate for the development of a modulator for both purposes. The effects of modulating food intake by rubiscolin-6 have been indicated with a rather lower dosage than that used in studies on other disease states [7,8,11]. The orexigenic effect is expected to stimulate food intake in patients with cancer who lose appetite due to cachexia [53] and elderly individuals with sarcopenia based on poor nutritional status, which contributes to a loss of muscle mass [54,55]. Other endogenous neuropeptides, such as neuropeptide Y, ghrelin, and orexin, have orexigenic activity but are inactivated when administered orally [56,57,58]; however, p.o. rubiscolin-6 has been shown to evoke these effects. Given that rubiscolin-6 is derived from RuBisCO, which is claimed to be the most abundant protein on earth [59], it might essentially be a drug of natural origin for humans and herbivores to regulate food intake. On the contrary, the anorexigenic effect of rubiscolin-6 will be useful during the current global trend of increasing obesity [60]. Although weight loss is the most effective intervention for reducing the risk of non-communicable diseases, such as type-2 diabetes and cardiovascular diseases, some patients still need pharmacotherapy. However, the existing pharmacotherapy is not always sufficient because of the weak effect or concerns for adverse effects [61]. Rubiscolin-6 is a unique opioid in that it suppresses high-fat intake via the activation of central DOR in mice fed a high-fat diet [12,13]. As a medication or supplementation with a novel mechanism of action, rubiscolin-6 might be valuable for patients who are obese.

### 3.2. Effects of Non-Orally Administered Rubiscolin-6

Besides the antidepressant-like effect of intraperitoneal administration indicated above, there are a couple of other effects indicated by in vitro studies or in vivo studies using the non-oral administration of rubiscolin-6. Chajra et al., conducted in vitro and in vivo studies (with aged human volunteers) and reported that DOR can be a modulator of skin differentiation and that rubiscolin-6 repaired skin damage by decreasing trans-epidermal water loss, hydration, and wrinkle depth at the periocular and perilabial areas [62]. The mechanism was speculated to involve inflammation inhibition induced by cytokines under stress conditions. Kairupan et al., reported the possibility of rubiscolin-6 improving impaired glucose uptake-related medical conditions, such as diabetes mellitus, as it increased glucose uptake potentially via the activation of AMP-activated protein kinase to enhance glucose transporter 4 translocation in skeletal muscle in streptozotocin-induced diabetic rats [63]. Considering the increasing number of cases of diabetes mellitus worldwide, this effect is also attractive. If this effect can be achieved via oral administration, expectations for developing rubiscolin-6 as a drug will further increase.

### 3.3. Safety of Rubiscolin-6

The safety of rubiscolin-6 is also an important aspect of drug development. It is considered to be relatively safe since it is derived from food. We clarified that rubiscolin-6 rarely recruited β-arrestin in in vitro studies, suggesting that unwanted effects, such as convulsion, which is sometimes associated with the administration of DOR agonists [13], or an increase in alcohol intake that correlates with β-arrestin recruitment induced by DOR agonists [64], are less likely to occur. It was reported that rubiscolin-6 had no effect on locomotor activity in a study investigating the anxiolytic effect [11]. However, only a few in vivo studies have been conducted on the safety of rubiscolin-6. Therefore, further animal studies and clinical studies are required to confirm its safety, especially the absence of induction of convulsion, which is the most concerning adverse effect that has historically been an obstacle to the development of DOR agonists as drugs [65].

### 3.4. Hypothesis for the Pharmacokinetics of Rubiscolins

Regarding pharmacokinetics, we present a hypothesis for the in vivo transportation of rubiscolin-6, as shown in Figure 1. Considering that rubiscolin-6 is more potent than rubiscolin-5, six amino acid residues with Phe in the sixth position are essential, and we hypothesize that rubiscolin-6 is not always degraded into dipeptides or tripeptides, thus resulting in oral availability. Some rubiscolin-6 molecules are transported as-is via the oligopeptide transporter SOPT-2 and flow into the bloodstream. The YP-sequence of rubiscolins includes Pro (a cyclic structured amino acid) in the second position, which is thought to confer stability to the peptide against endogenous peptidases and prevent it from being easily degraded into dipeptides or tripeptides [9,66]. To better understand the mechanism of absorption, further research is needed to investigate whether rubiscolin-6 is transported by SOPT-2. In addition, given that every reported effect observed with orally administered rubiscolin-6 was blocked by i.c.v. administered naltrindole, rubiscolin-6 must penetrate the BBB to exert the effect via the DOR pathway in the CNS. Furthermore, transporters, such as PTS-1, may be involved. Transportation is not merely determined by the number of amino acid residues of the peptide; the size, structure, polarity, charge, and protein binding rate of peptides are also considered [28]. As β-casomorphin, a milk-derived opioid heptapeptide (Tyr-Pro-Phe-Pro-Gly-Pro-Ile) [67], has similar properties to those of rubiscolin-6, such as molecular weight (β-casomorphin: 789.9 g/mol, rubiscolin-6: 766.9 g/mol) and polarity, we suppose that rubiscolin-6 may also be transported via the same or similar pathway as β-casomorphin. However, the similarity between these two peptides is limited. The hypothesis regarding the two pathways, that is, absorption in the intestinal tract and transportation across the BBB, should be further clarified.

### 3.5. Analogs or Derivatives of Rubiscolin-6

Analogs or derivatives of rubiscolin-6 have also been investigated to detect more advantageous synthetic peptides that are more potent than rubiscolin-6 itself [9]. The first two residues Tyr-Pro in the N-terminal sequence are not modified because they are essential for the opioid activity. Yang et al., evaluated the activities and receptor affinities of analogs of rubiscolin-6 using mouse vas deferens and guinea pig ileum assays after replacing specific amino acid residues. They reported that substituting the third Leu with Ile or Met increased the potency of DOR activity four times and that substituting the sixth Phe with Val could increase it by 10 times. In the research, YPMDLV was the most ideal derivative, which was nearly 20 times more potent than rubiscolin-6 and exhibited a more potent antinociceptive effect by i.c.v administration in ddY mice [9]. Caballero et al., conducted three-dimensional quantitative structure–activity relationship studies to investigate the structural requirements of rubiscolin-6 analogs for high DOR activity using comparative molecular field analysis and comparative molecular similarity index analysis [68]. They reported that Asp at the fourth residue is important based on the hydrophobic map because substitution with longer or shorter residues caused a marked decrease in the DOR activity. Regarding the fifth and sixth positions, their analyses indicated that hydrophobic residues are preferable for increasing the DOR activity. Marinaccio et al., reported that the synthetic analog YPMDIV was the most potent for increasing DOR activity, and they designed 12 new analogs using YPMDIV (Tyr-Pro-Met-Glu-Ile-Val-OH) as a lead compound [69]. They concluded that peptide groups with the sequences Tyr-Pro-Cys-Glu-Ile-Val-NH_2_, Tyr-d-Pro-Cys-Glu-Ile-Val-NH_2_, Tyr-d-Pro-Cys-Glu-Ile-Val-OH, and Tyr-Pro-Cys-Glu-Ile-Val-OH had the highest efficacy in terms of antinociception and anti-inflammation activities in in vivo studies via i.c.v. administration and the peptide Tyr-d-Pro-Cys-Glu-Ile-Val-NH_2_ exhibited the most potent antioxidant and tyrosinase-inhibitory activities in in vitro studies. Additionally, Tyr-Pro-Cys-Glu-Ile-Val-NH_2_ and Tyr-d-Pro-Cys-Glu-Ile-Val-OH were considered to have the potential to be developed as new nutraceuticals with cognitive-enhancing properties based on their anti-cholinergic effect [69]. Nevertheless, further in vivo investigations by oral administration will be required to identify more promising analogs or derivatives of rubiscolin-6.

## 4. Conclusions and Future Perspective

In conclusion, we believe that rubiscolin-6, a G protein-biased DOR-specific agonist, is an ideal candidate for the development of a novel and safe oral peptide drug for treating several diseases and symptoms. Further studies are required to elucidate the pharmacokinetics for the metabolic stability and BBB permeability, and the in vivo safety of rubiscolin-6 and its potential analogs.

## Figures and Tables

**Figure 1 ijms-24-09959-f001:**
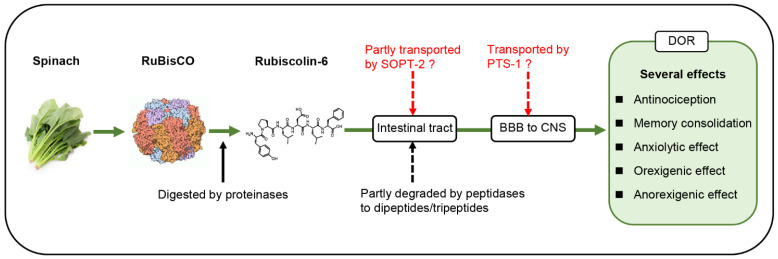
Hypothesis for in vivo transportation of rubiscolin-6. RuBisCO, D-ribulose-1,5-bisphosphate carboxylase/oxygenase; SOPT-2, sodium-coupled oligopeptide transporter-2; PTS-1, peptide transport system-1; BBB, blood–brain barrier; CNS, central nervous system; DOR, delta opioid receptor. Image of RuBisCO: Molecule of the Month ^©^ David S. Goodsell and RCSB PDB licensed under CC-BY-4.0 International.

**Table 1 ijms-24-09959-t001:** Summary of evidence of the effects of orally administered rubiscolins.

	Effect	ID	Animal	Effective Dose	Findings
1	Antinociceptive	Yang et al. [7]	Male ddY mice	Rubiscolin-5: 300 mg/kg, rubiscolin-6: 100 mg/kg	Both peptides exhibited an antinociceptive effect and were blocked by a selective δ opioid antagonist, naltrindole.
2	Memory-enhancing	Yang et al. [8]	Male ddY mice	Rubiscolin-6: 100 mg/kg	Rubiscolin-6 showed memory-enhancing effects; it was blocked by naltrindole. Notably, rubiscolin-5 failed to elicit this effect, even at 300 mg/kg.
3	Anxiolytic	Hirata et al. [11]	Male ddY mice	Rubiscolin-6: 100 mg/kg	Rubiscolin-6 exerted an anxiolytic effect and was blocked by naltrindole. The effect was mediated by σ1 and dopamine D1 receptors.
4	Orexigenic	Kaneko et al. [12]	Male ddY or C57BL/6 mice	Rubiscolin-6: 0.3 mg/kg	Rubiscolin-6 stimulated food intake and was blocked by naltrindole. The effect was mediated by cyclooxygenase-2 and lipocalin-type PGDS.
		Miyazaki et al. [14]	Male C57BL/6N mice	Rubiscolin-6: 1 mg/kg	Rubiscolin-6 stimulated food intake even in aged mice with ghrelin resistance and was blocked by naltrindole.
5	Anorexigenic	Kaneko et al. [13]	Male ddY or C57BL/6 mice	Rubiscolin-6: 0.3 mg/kg	Rubiscolin-6 suppressed food intake in mice on a high-fat diet and was blocked by naltrindole and HS024.

## Data Availability

Not applicable.

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
