# Peer review of "Therapeutic Potential of Orally Administered Rubiscolin-6"

_ijms, 2023, doi:10.3390/ijms24129959_

Round 1

Reviewer 1 Report

The topic of the review manuscript by Karasawa et al. is of significance in the contemporary context of opioid epidemic and the high need for safer pain therapies. The manuscript is a comprehensive and up-to-date review with a critical view of the research area. It centers on the current state of knowledge on the preclinical pharmacology of rubiscolin-6 after oral administration.

There are some specific points to be addressed by the authors.

In the introduction, the authors should better emphasize the advantages of peptides vs small molecules in drug development. In addition, a section on “Conclusions and future perspective” should be added.

The following references may be very relevant to be cited in the Introduction:

Line 43, when refereeing to novel drug development based on natural and chemical peptides: DOI: 10.3390/biom12091241 and doi: 10.1007/164_2021_519.

Line 63, when describing the role of the DOR: doi: 10.3390/ph15070873.

Line 21: revise ‘delta opioid receptor’ for ‘delta-opioid receptor’.

Line 65: revise ‘µ opioid receptor’ for ‘mu-opioid receptor’.

Line 80: revise ‘κOR’ for ‘kappa-opioid receptor’

Author Response

Responses to Reviewer Comments

Reviewer 1 comments:

The topic of the review manuscript by Karasawa et al. is of significance in the contemporary context of opioid epidemic and the high need for safer pain therapies. The manuscript is a comprehensive and up-to-date review with a critical view of the research area. It centers on the current state of knowledge on the preclinical pharmacology of rubiscolin-6 after oral administration.

There are some specific points to be addressed by the authors.

  1. In the introduction, the authors should better emphasize the advantages of peptides vs small molecules in drug development. In addition, a section on “Conclusions and future perspective” should be added.

Author response: According to your suggestion, we added “The approaches for developing a peptide drug are based on the expectation that peptides will be more effective, considering their higher specificity and lower toxicity than small molecules, which are broadly distributed in the body.” in the Introduction (page 1, lines 43–46). Moreover, the “Conclusion and future perspective” section has been added (page 9, lines 377–382).

  1. The following references may be very relevant to be cited in the Introduction:
  • Line 43, when refereeing to novel drug development based on natural and chemical peptides: DOI: 10.3390/biom12091241 and doi: 10.1007/164_2021_519.

Author response: Thank you for suggesting references related to the Introduction of our manuscript. We have cited these references as #2 and #3 in the Introduction (page 1, line 43).

  • Line 63, when describing the role of the DOR: doi: 10.3390/ph15070873.

Author response: Thank you for your suggestion. We cited the reference as #15 in the Introduction (page 2, line 67).

  • Line 21: revise ‘delta opioid receptor’ for ‘delta-opioid receptor’.

Author response: We have revised it as suggested (page 1, line 21).

  • Line 65: revise ‘µ opioid receptor’ for ‘mu-opioid receptor’.

Author response: We have revised it as suggested (page 2, line 68).

  • Line 80: revise ‘κOR’ for ‘kappa-opioid receptor’

Author response: We have revised it as suggested (page 2, line 83).

Reviewer 2 Report

The manuscript “Therapeutic potential of orally administered rubiscolin-6” presents the current state of knowledge on biological activity of rubiscolin 6. This topic gains a lot of attention, with several review papers being published, centered on food chemistry, with therapeutic potential as additional benefit.

The Authors concentrate on oral administration effects, however, other administration methods are mentioned and discussed, thus making the statement in Abstract (line 25/26) not completely representative.

The oral availability of rubiscolin is mentioned in several places (for example, line 75), however, no specific data on proteolytic stability are provided. There is no discussion on reasons for oral availability of this linear peptide. It is an unique feature of peptide and more information would be really beneficial. Some references to the material in lines 135-144 are required.

In line 140, BBB crossing mechanism is stated as unknown, then, in 158, “rubiscolin-6 enters the CNS via the BBB [21].” – the reference describes general mechanisms of opioids (?), however, this creates the impression that the mechanism for rubiscolin is known and reported in reference 21, please reconsider.

Structures of rubiscolins appear in figure, practically at the end of the text. Several agonists and antagonists of opioid receptors are mentioned, without structures. Therefore the only solution for people not specializing in opioid substances is to look for these structures on their own, if they want to compare the size and structure. It is especially important in the case of exomorphin A5

I would suggest dividing Discussion into subparagraphs with titles related to described activity, to make reading more focused. The part describing the non-oral administration would also benefit for a separate paragraph title.

In line 293, there is a statement “However, only a few in vivo studies have been conducted on the safety of rubiscolin-6”, concerning very important properties of rubiscolin. More information is needed (apart from mentioned lack of convulsions) as well as more references.

Line 301: the mentioned hypothesis, without comment on literature data (or lack of studies) is not serious. The comment in Figure 1 (partly degraded….) promises extremely interesting activity if a more stable form could be obtained – however, the level of degradation is crucial.
(for example, Stefanucci 2020 doi.org/10.1016/j.jff.2020.104154)

Line 313: the comparison of β-casomorphin (casomorphin-7, one of the family of several peptides, Tyr-Pro-Phe-Pro-Gly-Pro-Ile) and rubiscolin 6 (Tyr-Pro-Leu-Asp-Leu-Phe) in this fragment is unjustified. The importance of characteristic casomorphin sequence is completely lost.

Minor comments:

line 56: penta- or hexa-amino acid residues – it is either penta- hexapeptide of five- six amino acid residues

line 58 (and other places in the text)  - the “d” in d-ribulose-1,5-bisphosphate should be replaced with “D”, font size reduced (D-ribulose…)

D in D-amino acids also requires  reduced font size.

Table 1, orexigenic – “or” after male ddY is located in confusing place

Line 263: In the sentence “Given that rubiscolin-6 is derived from the most abundant protein on earth found in green leaves, it might essentially be a drug of natural origin for humans and herbivores to regulate food intake.” it is not clear whether Rubisco (?) is the most abundant protein on earth or most abundant protein on earth found in green leaves (in green leaves only). Please clarify and provide reference.

Line 332: Lue? Leu? please verify

Line 350 and following: the use of “pro” in original paper is quite specific, to accompany the figure, as both isomers were synthesized and tested (separately). Please indicate the correct chirality of Pro residue to present a consistent nomenclature scheme in this manuscript.

Author Response

Reviewer 2 comments:

The manuscript “Therapeutic potential of orally administered rubiscolin-6” presents the current state of knowledge on biological activity of rubiscolin 6. This topic gains a lot of attention, with several review papers being published, centered on food chemistry, with therapeutic potential as additional benefit.

  1. The Authors concentrate on oral administration effects, however, other administration methods are mentioned and discussed, thus making the statement in Abstract (line 25/26) not completely representative.

Author response: We have revised the sentence as “In this review, we show the therapeutic potential of rubiscolin-6, mainly focusing on its effects when orally administered, based on available evidence.” in the Abstract (page 1, lines 25–26), according to your suggestion.

  1. The oral availability of rubiscolin is mentioned in several places (for example, line 75), however, no specific data on proteolytic stability are provided. There is no discussion on reasons for oral availability of this linear peptide. It is an unique feature of peptide and more information would be really beneficial. Some references to the material in lines 135-144 are required.

Author response: We have added a sentence with a related reference as “Stefanucci et al. investigated the in vitro intestinal stability and bioavailability of rubiscolin-6 using single layers of CaCo2 cells as a model of absorption in the small intestine [20]. They reported that approximately 10% of rubiscolin-6 was transepithelially transported.” (page 4, lines 142–145).

  1. In line 140, BBB crossing mechanism is stated as unknown, then, in 158, “rubiscolin-6 enters the CNS via the BBB [21].” – the reference describes general mechanisms of opioids (?), however, this creates the impression that the mechanism for rubiscolin is known and reported in reference 21, please reconsider.

Author response: As we described in the manuscript, there is scanty research on how rubiscolin-6 enters and penetrates the BBB. To avoid such misunderstanding, we have revised the sentence as “Regarding the second mechanism, rubiscolin-6 has been considered to enter the CNS via the BBB, although the mechanism of opioid peptide drug delivery to the CNS remains unclear”, because the reference is a general review for CNS drug delivery and opioid peptides (page 5, lines 164–166).

  1. Structures of rubiscolins appear in figure, practically at the end of the text. Several agonists and antagonists of opioid receptors are mentioned, without structures. Therefore the only solution for people not specializing in opioid substances is to look for these structures on their own, if they want to compare the size and structure. It is especially important in the case of exomorphin A5.

Author response: Thank you for your suggestion. We have added the amino acid sequence of each peptide at the first mention in the manuscript (page 5, lines 171–174, page 6, line 230, page 8, line 333) as well as the molecular weight of β-casomorphin and rubiscolin-6 (page 8, lines 334–335), accordingly.

  1. I would suggest dividing Discussion into subparagraphs with titles related to described activity, to make reading more focused. The part describing the non-oral administration would also benefit for a separate paragraph title.

Author response: As suggested, we have divided the Discussion into the following subsections: “3.1. Therapeutic potential of orally administrated rubiscolin-6” (page 5, line 183), “3.2. Effects of non-orally administrated rubiscolin-6” (page 7, line 287), “3.3. Safety of rubiscolin-6” (page 7, line 303), “3.4. Hypothesis for the pharmacokinetics of rubiscolins” (page 8, line 316), and “3.5. Analogs or derivatives of rubiscolin-6” (page 8, line 346), according to your suggestion.

  1. In line 293, there is a statement “However, only a few in vivo studies have been conducted on the safety of rubiscolin-6”, concerning very important properties of rubiscolin. More information is needed (apart from mentioned lack of convulsions) as well as more references.

Author response: Although we have comprehensively and thoroughly reviewed the in vivo studies on rubiscolin-6, safety data are limited among all studies. Nevertheless, we have added the sentences “It is considered to be relatively safe since it is derived from food.” (page 7, lines 304–305), and “It was reported that rubiscolin-6 had no effect on locomotor activity in a study investigating the anxiolytic effect [11]” (page 7, lines 309–310).

  1. Line 301: the mentioned hypothesis, without comment on literature data (or lack of studies) is not serious. The comment in Figure 1 (partly degraded….) promises extremely interesting activity if a more stable form could be obtained – however, the level of degradation is crucial.
    (for example, Stefanucci 2020 doi.org/10.1016/j.jff.2020.104154)

Author response: Thank you for your suggestion and providing the related reference. We have added a sentence for the possible level of degradation: “ Stefanucci et al. investigated the in vitro intestinal stability and bioavailability of rubiscolin-6 using single layers of CaCo2 cells as a model of absorption in the small intestine [20]. They reported that approximately 10% of rubiscolin-6 was transepithelially transported” (page 4, lines 142–145).

  1. Line 313: the comparison of β-casomorphin (casomorphin-7, one of the family of several peptides, Tyr-Pro-Phe-Pro-Gly-Pro-Ile) and rubiscolin 6 (Tyr-Pro-Leu-Asp-Leu-Phe) in this fragment is unjustified. The importance of characteristic casomorphin sequence is completely lost.

Author response: To avoid misunderstanding and to indicate our thoughts, we have revised the sentence as “As β-casomorphin, a milk-derived opioid heptapeptide (Tyr-Pro-Phe-Pro-Gly-Pro-Ile), has similar properties to those of rubiscolin-6, such as molecular weight (β-casomorphin: 789.9 g/mol, rubiscolin-6: 766.9 g/mol), polarity, and charge, we suppose that rubiscolin-6 may also be transported via the same pathway as β-casomorphin.” (page 8, lines 332–336)

Minor comments:

  1. line 56: penta- or hexa-amino acid residues – it is either penta- hexapeptide of five- six amino acid residues

Author response: We have revised it as “penta- or hexapeptide”, according to your comment (page 2, line 60).

  1. line 58 (and other places in the text)  - the “d” in d-ribulose-1,5-bisphosphate should be replaced with “D”, font size reduced (D-ribulose…)

Author response: We have revised it with the appropriate font (page 2, line 62).

  1. D in D-amino acids also requires  reduced font size.

Author response: We have revised them with the appropriate font (page 9, lines 367, 369 and 371).

  1. Table 1, orexigenic – “or” after male ddY is located in confusing place

Author response: We have adjusted the style of Table 1 to avoid confusion (Table 1).

  1. Line 263: In the sentence “Given that rubiscolin-6 is derived from the most abundant protein on earth found in green leaves, it might essentially be a drug of natural origin for humans and herbivores to regulate food intake.” it is not clear whether Rubisco (?) is the most abundant protein on earth or most abundant protein on earth found in green leaves (in green leaves only). Please clarify and provide reference.

Author response: We have deleted “found in green leaves” and revised the sentence as “Given that rubiscolin-6 is derived from RuBisCO, which is claimed to be the most abundant protein on earth [59], it might essentially be a drug of natural origin for humans and herbivores to regulate food intake.” (page 7, lines 274–277).

  1. Line 332: Lue? Leu? please verify

Author response: We have corrected it as “Leu” (page 8, line 352).

  1. Line 350 and following: the use of “pro” in original paper is quite specific, to accompany the figure, as both isomers were synthesized and tested (separately). Please indicate the correct chirality of Pro residue to present a consistent nomenclature scheme in this manuscript.

Author response: We have specified the chirality of tested peptides according to the reference (page 9, lines 367, 369 and 371).

Round 2

Reviewer 2 Report

The changes in the manuscript solved most of the indicated problems and the text is now easier to fully appreciate. 

The question on stability was not answered directly. The 10% transport was described as interesting in the reference, whereas the hydrolysis was quite moderate, less than 10% (90% remaining). Stefanucci et al. also indicates that the Rubiscolin-6 amide is much more susceptible for hydrolysis (74%).
Such difference for short linear peptide is interesting.

Please check the casomorphin sequence (line 347) as it contains only 6 amino acid residues, whereas it is described as heptapeptide.
The charge of rubiscolin (at pH 7) would be different than the charge of this casomorphin, because of aspartic acid residue (Asp), therefore the similarity is limited.

Author Response

June 7, 2023

Dear Reviewer,

Thank you for providing the reports on our manuscript, " Therapeutic potential of orally administered rubiscolin-6". Based on the comments received, our manuscript has been revised (in track changes) and, below, we submit a list of responses to the comments.

Kind regards,

Yusuke Karasawa, Ph.D.

The changes in the manuscript solved most of the indicated problems and the text is now easier to fully appreciate. 

The question on stability was not answered directly. The 10% transport was described as interesting in the reference, whereas the hydrolysis was quite moderate, less than 10% (90% remaining). Stefanucci et al. also indicates that the Rubiscolin-6 amide is much more susceptible for hydrolysis (74%). Such difference for short linear peptide is interesting.

Author response:

Thank you for your comment. We’re afraid that we do not have the direct answer to it yet. To answer the important question, we hope further research could elucidate the profiles of rubiscolin-6 stability.

Please check the casomorphin sequence (line 347) as it contains only 6 amino acid residues, whereas it is described as heptapeptide.

Author response:

Thank you for your point and we have revised the sequence of casomorphin as “Tyr-Pro-Phe-Pro-Gly-Pro-Ile”, since we had missed to describe the fourth Proline in it. (page 5, line 172 and page8, line 334)

The charge of rubiscolin (at pH 7) would be different than the charge of this casomorphin, because of aspartic acid residue (Asp), therefore the similarity is limited.

Author response:

According to your suggestion, we have deleted the “charge” from the sentence (page8, line 334). In addition, we have added the sentence “However, the similarity between these two peptides is limited.” (page 8, line 337-338).